# Alcalase-Assisted *Mytilus edulis* Hydrolysate: A Nutritional Approach for Recovery from Muscle Atrophy

**DOI:** 10.3390/md21120623

**Published:** 2023-11-29

**Authors:** R. P. G. S. K. Amarasiri, Jimin Hyun, Sang-Woon Lee, Jin Kim, You-Jin Jeon, Jung-Suck Lee

**Affiliations:** 1Department of Marine Life Science, Jeju National University, Jeju 63243, Republic of Korea; sewwandi1995@office.jejunu.ac.kr (R.P.G.S.K.A.); localman@office.jejunu.ac.kr (J.H.); pucu2195@gmail.com (S.-W.L.); 2Department of Seafood and Aquatic Science, Gyeongsang National University, Tongyeong 53064, Republic of Korea; kimjin501@gnu.ac.kr

**Keywords:** Alcalase, blue mussel, marine bioresource, muscle atrophy, *Mytilus edulis*, nutritional supplementation

## Abstract

Muscle atrophy is a complex physiological condition caused by a variety of reasons, including muscle disuse, aging, malnutrition, chronic diseases, immobilization, and hormonal imbalance. Beyond its effect on physical appearance, this condition significantly reduces the quality of human life, thus warranting the development of preventive strategies. Although exercising is effective in managing this condition, it is applicable only for individuals who can engage in physical activities and are not bedridden. A combination of exercise and nutritional supplementation has emerged as a more advantageous approach. Here, we evaluated the effects of enzyme-assisted hydrolysates of *Mytilus edulis* prepared using Protamex (PMH), Alcalase (AMH), or Flavourzyme (FMH) in protecting against muscle atrophy in a dexamethasone (Dex)-induced muscular atrophy model in vitro and in vitro. Alcalase-assisted *M. edulis* hydrolysate (AMH) was the most efficient among the tested treatments and resulted in higher protein recovery (57.06 ± 0.42%) and abundant amino acid composition (43,158 mg/100 g; 43.16%). AMH treatment also escalated the proliferation of C2C12 cells while increasing the total number of nuclei, myotube coverage, and myotube diameter. These results were corroborated by a successful reduction in the levels of proteins responsible for muscle atrophy, including E3 ubiquitin ligases, and an increase in the expression of proteins associated with muscle hypertrophy, including myogenin and MyHC. These results were further solidified by the successful enhancement of locomotor ability and body weight in zebrafish following AMH treatment. Thus, these findings highlight the potential of AMH in recovery from muscle atrophy.

## 1. Introduction

Skeletal muscle is the most abundant tissue in the human body, accounting for approximately 40% of the total body weight, and plays a critical role in maintaining the overall health and functionality of an individual. The remarkable capacity of skeletal muscles for contraction and relaxation enables crucial body movements, such as locomotion, respiration, and posture control [1]. Furthermore, skeletal muscles serve as a reservoir for nutrients, such as proteins and glucose, regulates energy expenditure, and influences metabolic homeostasis [2].

Muscle atrophy is a complex and multifactorial condition that involves the loss of muscle mass and strength due to an impaired balance between protein synthesis and degradation. This debilitating condition can result from a variety of factors, including prolonged immobilization; malnutrition; and specific pathological conditions, such as sarcopenia and cachexia [3,4]. Sarcopenia refers to a specific type of muscle atrophy that is primarily associated with aging, whereas cachexia is a syndrome characterized by weight loss; muscle wasting; and systemic inflammation that is commonly observed in individuals with chronic diseases, such as cancer, heart failure, and AIDS [5,6]. These detrimental conditions considerably affect the quality of human life as they result in decreased physical performance, reduced functional independence, and overall impaired well-being. Although exercise interventions are commonly recommended and effective in managing these conditions, their applicability is primarily limited to individuals who can engage in physical activities and are not bedridden. Moreover, exercise alone may not be sufficient to prevent muscle atrophy caused by sarcopenia and cachexia wherein chronic conditions or aging-related factors contribute to muscle loss. Therefore, a combination of adequate dietary protein, energy, and progressive resistance exercise has emerged as a more effective approach than exercise alone [7]. Consequently, preventative measures that can effectively counteract muscle atrophy and mitigate its adverse effects are crucial. Dietary supplements derived from natural sources provide a potential approach to address muscle atrophy owing to their ability to modulate various physiological processes involved in the growth and maintenance of muscles [8].

*Mytilus edulis*, commonly known as the blue mussel, holds significant economic and nutritional value, which makes it a highly valued species in the commercial and culinary domains. In 2016, global mussel production reached 2.14 million tons, reflecting a significant growth rate of 35% in total production in 2007. China is the largest producer of mussels globally, achieving an output of 879,000 tons in 2016, whereas production in the Republic of Korea was 64,000 tons in that year [9]. The popularity of mussels differs across countries, with per capita consumption ranging from less than 0.2 kg to almost 4.0 kg. Of the 500 million individuals that make up the European population, approximately 200 million consume 2.60 kg mussels per annum on average, whereas the remaining 300 million consume lesser amounts [10]. This marine bivalve mollusk holds a crucial position in the human diet, being a highly nutritious seafood option that is a rich source of protein, essential amino acids, vitamins (B_12_ and D), minerals (F, Zn, and Se), and omega-3 fatty acids [11]. Furthermore, it offers a variety of health benefits owing to its high nutritional value and the presence of bioactive compounds [12]. Potent bioactivities of *M. edulis*, including antihypertension, antioxidant, antibacterial, anticoagulant, anti-inflammatory, osteogenic, hepatoprotective, and anticancer properties, have been reported [13,14,15,16,17].

In recent years, there has been a growing demand for functional foods that offer health benefits beyond basic nutrition [18]. Among the available functional foods, those specifically targeting muscle growth and health are relatively scarce. This presents an opportunity to explore the potential of *Mytilus edulis* as a functional food ingredient for this purpose. Other mussel species, such as the New Zealand green-lipped mussel (*Perna canaliculus*), are sold as functional food in many countries. For instance, nutritional supplements like Seatone, containing freeze-dried tissue powder from the mussel, are used to relieve arthritic symptoms and to aid the regeneration of arthritic and injured joints [19]. This precedent suggests that *Mytilus edulis*, with its high protein content and diverse bioactivities, could similarly be harnessed for its potential benefits, including those related to muscle growth. Nevertheless, the use of crude raw material as functional food is fraught with several concerns that need to be addressed. These include safety hazards due to the presence of undesirable substances, low product quality resulting from impurities and inconsistencies, unfavorable aroma and taste, and digestibility. Enzyme-assisted extraction of bioactive compounds from the source material offers significant benefits, including efficient extraction resulting in higher yields and enhanced digestibility that renders these compounds more readily absorbable by the body. Additionally, enzymes eliminate unfavorable aroma and taste often associated with crude extracts, thereby improving the sensory appeal of the final product. These advantages highlight the potential of enzyme-assisted extraction as a preferred method for producing functional food with improved quality, digestibility, and sensory characteristics. In this study, we investigated the potential of enzyme-assisted hydrolysates of *M. edulis* (EMHs) to attenuate muscle atrophy using in vitro and in vivo zebrafish models.

## 2. Results

### 2.1. Proximate Composition Analysis

To assess the industrial applicability of *M. edulis*, enzymatic hydrolysis was performed with minor modifications from a previously described method, and its hydrolysis efficacy was determined (Figure 1 and Table 1) [20]. The hydrolysate yields of DW, PMH, AMH, and FMH were 44.40 ± 1.60%, 72.90 ± 2.90%, 76.37 ± 0.77%, and 63.22 ± 2.02%, respectively (Table 1). The hydrolysate yield was significantly higher in enzyme-assisted extracts compared to that in DW. The protein content was also similar, with the DW having the lowest protein recovery (43.61 ± 1.36%) and EMHs showing significantly higher protein recovery (PMH: 56.18 ± 0.77%, AMH: 57.33 ± 2.40%, and FMH: 51.95 ± 0.75%). In contrast, the lipid content of the enzyme-assisted extracts did not significantly differ from that of the DW extract (DW: 5.86 ± 0.56%, PMH: 6.48 ± 0.18%, AMH: 6.62 ± 0.03%, and FMH: 6.19 ± 0.17%). Thus, enzyme-assisted hydrolysis (PMH, AMH, and FMH) outperformed DW hydrolysis with significantly higher hydrolysate yield and protein content.

### 2.2. Effect of EMHs on the Viability, Proliferation, and Dexamethasone-Induced Proliferation of C2C12

The cytotoxic effect of DW, PMH, AMH, and FMH on C2C12 cells was tested. No cytotoxicity was observed at the tested concentrations (10, 30, 100, and 300 µg/mL; Figure 2a). Therefore, these concentrations were used for further experiments. To evaluate the effect on the proliferation of C2C12s, we performed the BrdU assay. The cells were treated with and without dexamethasone (Dex) (10 µM) stimulation; 100 µg/mL maca was used as a positive control. PMH and AMH treatments significantly increased the proliferation of C2C12 cells. In contrast, DW and FMH treatment did not significantly increase the proliferation of C2C12 cells compared to the control treatment (Figure 2b). Dex stimulation significantly reduced the number of BrdU-positive cells compared to the control, whereas DW, PMH, AMH, and FMH treatments showed a significant cell-promoting effect on Dex-stimulated C2C12 cells compared to the Dex group (Figure 2c). The maximum cell proliferation was detected for the 300 µg/mL AMH treatment. Thus, among the tested EMHs, AMH exhibited an excellent cell proliferative effect compared to the control.

### 2.3. Effect of EMHs on Differentiation of C2C12 Myotubes

The effects of EMHs on the total number of nuclei, myotube coverage, and myotube diameter in differentiated C2C12 myotubes were assessed using immunofluorescence employing the MyHC antibody. Figure 3a shows the immunofluorescence images of differentiated C2C12 myotubes subjected to DW treatment. DW treatment significantly increased the total number of nuclei (Figure 3b), myotube coverage up to 30 µg/mL (Figure 3c), and myotube diameter (Figure 3d) compared to the Dex group but did not show a dose dependence. Figure 4a shows the immunofluorescence images after PMH treatment, which considerably increased the total number of nuclei (Figure 4b), myotube coverage (Figure 4c), and myotube diameter (Figure 4d) compared to the Dex group; however, no dose dependence was noted. Immunofluorescence images of AMH-treated myotubes (Figure 5a) showed a drastic increase in the total number of nuclei (Figure 5b), myotube coverage (Figure 5c), and myotube diameter (Figure 5d) compared to the Dex group. Moreover, the mean values of each parameter increased with increasing concentration. FMH-treated myotubes (Figure 6a) also exhibited an increase in the number of nuclei (Figure 6b), myotube coverage (Figure 6c), and myotube diameter (Figure 6d) compared to the Dex group. Overall, AMH demonstrated a significant effect on Dex-induced myocyte differentiation compared to the other EMHs.

### 2.4. Effect of AMH on the Expression of Proteins Responsible for Muscle Hypertrophy and Atrophy

The effect of AMH treatment on the expression of proteins involved in muscle atrophy and hypertrophy was evaluated using Western blot analysis. Figure 7a shows the expressions of MAFbx, Murf-1, MyHC, and myogenin after AMH treatment. The treatment significantly downregulated the expression of proteins responsible for muscle atrophy, including MAFbx (Figure 7b) and Murf-1 (Figure 7c), but upregulated the expression of proteins responsible for muscle hypertrophy, including MyHC (Figure 7d) and myogenin (Figure 7e).

### 2.5. Recovery of Dex-Induced Muscle Atrophy in Zebrafish by AMH Administration

To investigate the potential of AMH to mitigate Dex-induced muscle atrophy and facilitate recovery, a zebrafish model was employed to assess their ability to counteract the effects of Dex in vivo (Figure 8a). Dex treatment resulted in significant impairment of locomotor function, as evidenced by factors such as total distance moved (Figure 8c), velocity (Figure 8d), mobility state (Figure 8e), and immobility state (Figure 8f) in the Dex group. Furthermore, Dex treatment notably reduced the body weight compared to the control group (Figure 8). Incorporation of 3% maca into the fish diet substantially improved swimming performance, as indicated by the mentioned muscle strength indicators in zebrafish, namely, total distance moved (Figure 8b), velocity (Figure 8c), mobility state (Figure 8d), and immobility state (Figure 8e), and significantly enhanced the body weight compared to the Dex group (Figure 8f). Conversely, incorporating 3% AMH into the fish diet yielded remarkable improvements in these Dex-induced effects. Zebrafish treated with AMH exhibited a substantial enhancement in swimming performance, characterized by increased indicators of muscle strength, including an expanded total distance moved (Figure 8b), heightened swimming velocity (Figure 8c), and a shift towards a more mobile state (Figure 8d).

### 2.6. Amino Acid Composition of AMH36

The amino acid composition analysis of the AMH sample revealed a comprehensive profile of constituent amino acids. The results are summarized in Table 2, which provides the relative concentrations of essential and nonessential amino acids present in the sample. From the total amino acid profile, 46.73% represented essential amino acids and 53.27% were nonessential amino acids (Table 2). Moreover, negatively charged amino acid content, aromatic amino acid content, and branched-chain amino acid content were 24.42%, 7.92%, and 16.84%, respectively (Table 2). Lysine, leucine, and arginine exhibited the highest relative abundances among the essential amino acids, and glutamic acid was the most abundant amino acid (5883 mg/100 g) in the AMH extract (Table 2)

## 3. Discussion

Proteins play a vital role in the growth and development of muscles and are therefore important for individuals seeking to enhance their muscle mass and performance. Among different sources of protein, seafood is an excellent option owing to its unique nutritional profile and bioactive constituents. Seafood offers a high-quality protein content characterized by a complete profile of essential amino acids, which are crucial for muscle protein synthesis but cannot be synthesized by the body [21]. Seafood proteins are rich in essential amino acids, such as leucine, which significantly affect muscle protein synthesis and anabolic signaling pathways [22]. Thus, owing to the high-quality protein content, complete amino acid profile, high bioavailability, and favorable fatty acid composition, there is an increasing demand and interest toward seafood as a highly nutritious protein source. In this study, we investigated the potential of EMH in attenuating Dex-induced muscle atrophy using an in vitro cell model. Based on previous studies demonstrating the effect of maca (*Lepidium meyenii*) in mitigating muscle atrophy, we employed maca as the positive control [23,24]. Our results showed that AMH treatment effectively increased cell proliferation and myotube diameter, indicating its potential to counteract muscle atrophy. Considering its potential industrial applications, particularly in nutritional approaches and functional food development, we noted the versatile benefits of Alcalase enzyme, including broad substrate specificity, stability across a wide pH range, high degree of hydrolysis, and the resulting flavor-enhancing hydrolysates with excellent solubility [25,26,27,28,29,30,31,32]. Thus, we further investigated AMH using Western blot analysis, amino acids composition analysis, and in vitro study using a zebrafish model.

First, we performed proximate composition analysis of EMHs to determine their nutritional value. The hydrolysis yield of EMHs varied, with AMH exhibiting the highest hydrolysate yield compared to DW (Table 1). The protein composition was the highest among the components, and AMH had a relatively higher protein recovery rate (1.30 ± 0.04-fold increase compared to DW) (Table 1). Compared to previous data on the nutritional components of *M. edulis*, the protein recovery yield of AMH in this study was drastically higher (Table 1). Fernandez et al. (2015) determined the nutrition composition of *M. edulis* and reported that the protein content of raw meat ranged from 43.36% to 62.88% [33]. The protein content in the AMH appears to be the maximum yield obtainable from *M. edulis*, highlighting the significant industrial value of AMH (Table 1). Interestingly, high carbohydrate content was another major feature of EMHs, including AMH (Table 1). This was due to the abundant glycogen storage in *M. edulis*, with a reported range of 4.10–36.7% [34,35]. In this study, the polysaccharide content ranged from 21.61 ± 0.53% to 30.20 ± 1.38% (Table 1). High polysaccharide content in AMH may significantly impact muscle growth because polysaccharides are a readily available source of energy, which increase the cell volume and modulate hormonal responses and signaling pathways, thus facilitating muscle protein synthesis and overall muscle adaptation [36]. Glycogen is critical for muscle growth and recovery owing to its remarkable ability to replenish muscle energy stores and sustain stamina after prolonged physical activity [37,38]. Therefore, concurrent consumption of protein and carbohydrates aids rapid recovery of muscle damage and promotes muscle growth [39,40].

AMH promoted muscle cell proliferation, followed by enhanced myotube differentiation characterized by increased myotube diameter, myotube coverage, and total number of nuclei (Figure 2 and Figure 5). These effects of AMH can be discussed in the context of the phases of myogenesis. During the cell proliferation phase, AMH had a positive effect on muscle cell proliferation, as evidenced by the increase in the total number of nuclei, indicating an augmented pool of myoblasts available for subsequent differentiation (Figure 2 and Figure 5). The increased myotube diameter and myotube coverage indicate enhanced fusion of myoblasts into larger and more developed myotubes [41]. This suggests that AMH promotes the formation of mature muscle fibers, which is crucial for functional muscle tissues (Figure 5). Numerous studies have reported the potent bioactive properties of Alcalase-assisted hydrolysates derived from food proteins on muscle cell proliferation [42]. Kim et al. (2019) investigated the effect of various enzyme hydrolysates derived from big-belly seahorse (*Hippocampus abdominalis*) on skeletal muscle growth in C2C12 cells as well as in a zebrafish model [43]. Similar to our findings, they observed that the Alcalase-assisted hydrolysate exhibited the highest cell proliferation among the tested enzyme hydrolysates (Figure 2b). These bioactivities are ascribed to the endopeptidase characteristics of Alcalase, which enable the production of diverse bioactive peptides with significant biological activities [44]. Alcalase is a serine endopeptidase derived from *Bacillus licheniformis*, which exhibits broad catalytic activity and hydrolyzes the peptide bond in the interior of polypeptide chains [45]. Its ability to recognize a diverse range of amino acids, either alone or in combination with other proteases, provides higher yield of hydrolysate containing small-sized peptides [46]. Compared to exopeptidases, the selective cleavage by endopeptidases enables breakdown of proteins at multiple sites, which allows more extensive and nuanced breakdown of the protein structure and ultimately leads to a more diverse range of peptide fragments [46].

Consistent with these findings, we observed upregulation of myogenin and MyHC, which indicates the positive effect of AMH treatment on the key protein expressions involved in myotube differentiation and muscle fiber maturation (Figure 7). Besides promoting the expression of muscle-hypertrophy-related proteins, AMH also downregulated proteins associated with muscle atrophy, including MAFbx and MuRF-1 (Figure 7). These proteins are key regulators of the breakdown of muscle proteins through the ubiquitin–proteasome pathway [47]. The downregulation of these proteins by AMH suggests its potential in attenuating muscle protein degradation and in preserving muscle mass (Figure 7). Our results are consistent with previous findings on the effect of bioactive compounds on muscle cell differentiation. Hsieh et al. (2020) reported the effect of teaghrelin, an active ingredient of Chin-shin oolong tea, on murine C2C12 myoblasts. Similar to our results, the treatment increased the Dex-induced myotube diameter, upregulated myogenin and MyHC, and reduced the expression of E3 ubiquitin ligases [48].

In an in vivo zebrafish experiment, we utilized a dexamethasone-induced muscle atrophy model known for its potent induction of muscular atrophy, resulting in diminished swimming endurance and impaired motor function [23]. The detrimental impact of Dex treatment on locomotor function was evident in heatmap and tracking images (Figure 8b). Furthermore, zebrafish in the Dex group experienced a marked reduction in their swimming capabilities; in contrast, incorporating 3% AMH into the fish diet yielded remarkable improvements in exploring behavior, characterized by increased indicators of muscle strength. The favorable effects of AMH were further evident through a significant increase in body weight compared to the Dex-treated group (Figure 8g). Because muscles are one of the heaviest tissues in an organism, this observation indicates that the inclusion of AMH in the zebrafish diet not only enhanced their exploring behaviors but also contributed to their overall well-being by preventing Dex-induced loss of body weight [49]. These findings align with our previous study, which provided strong evidence for the potential benefits of olive flounder processing by-products in alleviating Dex-induced muscle atrophy in a zebrafish model, emphasizing its role in enhancing muscle health and functionality [50].

The impact of amino acids, particularly branched-chain amino acids (BCAAs), on muscle growth is well recognized due to their role in protein synthesis and overall muscle metabolism [51]. For instance, whey protein is renowned for its high BCAA content and rapid digestion, making it a favored protein source among athletes and bodybuilders [52]. BCAAs, including leucine, isoleucine, and valine, are essential for stimulating muscle protein synthesis, which is pivotal for muscle growth and repair. Leucine acts as a key trigger for the mTOR pathway, a cellular signaling mechanism that orchestrates protein synthesis [53]. AMH showed relatively high BCAA content, including high leucine content (3016 mg/100 g, 6.9% of total AAs), suggesting a potential positive impact on muscle protein synthesis and, consequently, muscle growth (Table 2).

In our HPLC analysis, we identified regular peaks in fraction 3, which was the most refined fraction among the three distinct fractions (Appendix A). This striking consistency in the elution behavior of these peaks suggests that there may be an existence of peptides with specific and reproducible characteristics, which holds particular significance for the observed activity in our study. The third fraction (Appendix A) can be subjected to further in-depth analyses to elucidate the identity of the peptide responsible for the restoration of the atrophied muscle. Therefore, additional studies are required to unveil the unknown substances in AMH.

Our results indicate that AMH treatment affects the muscle cells, influencing the proliferation, differentiation, and expression of proteins involved in muscle hypertrophy and atrophy. The increase in myotube diameter upon AMH treatment can be attributed to multiple factors. One of the factors could be that hydrolysates modulate signaling pathways, such as the PI3K/Akt/mTOR pathway, involved in muscle growth. Activation of this pathway promotes protein synthesis and muscle hypertrophy [54]. Thus, future studies should investigate the specific mechanisms by which AMH enhances myotube diameter and protein synthesis. Understanding the underlying mechanism could open new avenues for developing functional food and therapeutic strategies for conditions characterized by muscle loss, such as sarcopenia and other neuromuscular disorders. Moreover, the bioavailability and physiological response to these peptides within a living organism can be influenced by various factors, including digestion, absorption, metabolism, and other physiological processes [55]. Therefore, even though the in vitro and in vivo findings of our work provide valuable insights into the potential bioactive properties of the AMH extract, further in vivo studies and human clinical trials are necessary to validate these outcomes.

## 4. Conclusions

In conclusion, among the initially tested hydrolysates, including DW, PMH, AMH, and FMH, AMH demonstrates the potential in protecting against muscle atrophy. AMH treatment resulted in increased proliferation of C2C12 cells, increased myotube coverage and diameter, and reduced levels of proteins responsible for muscle atrophy. Moreover, the results were further solidified by remarkable improvements in exploring behavior and elevated body weight of the zebrafish compared to the Dex-group after the incorporation of 3% AMH into the diet in our in vivo study. These findings suggest that AMH could be a promising nutritional supplement for the prevention of and recovery from muscle atrophy. Further research is required to validate the identity of the compound through mass analysis for its commercial utilization. Additionally, the specific mechanisms by which AMH enhances the myotube diameter and protein synthesis should be investigated through animal experiments and clinical trials.

## 5. Materials and Methods

### 5.1. Reagents

The required chemicals and mixtures were procured from Sigma-Aldrich (St. Louis, MO, USA) unless otherwise indicated. Goat antimouse IgG H&L (Alexa Fluor^TM^ Plus 555; A32727, Thermo Fisher Scientific Inc., Waltham, MA, USA) for immunofluorescence was obtained from Invitrogen (Cambridge, UK). Primary and secondary antibodies for Western blot analysis were from Santa Cruz Biotechnology (Santa Cruz Biotechnology Inc., Dallas, TX 75220, USA). The three digestive proteases, Alcalase, Flavourzyme, and Protamex, were purchased from Novo Co. (Novozyme Nordisk, Bagsvaerd, Denmark). Commercially available condensed maca (*Lepidium meyenii*), which is used in food supplements, was purchased from NOW foods (Lot No. 3063322, Bloomingdale, IL, USA) and used as the positive control.

### 5.2. Cell Lines and Cell Culture

C2C12 myoblasts were purchased from American Type Culture Collection (ATCC, Manassas, VA, USA). Cells were cultured in Dulbecco’s modified Eagle’s medium (DMEM; WELGENE Inc., Gyeongsangbuk-do, Republic of Korea) containing 10% fetal bovine serum (FBS; WELGENE Inc.) and 1% penicillin and streptomycin (Invitrogen, Waltham, MA, USA) at 37 °C in a 5% CO_2_ humidified incubator (60915672, Sanyo Electric Co., Ltd., Osaka, Japan). At 70–80% confluence, cell differentiation was induced by replacing the culture medium with DMEM containing 2% horse serum (HS; Gibco, Invitrogen Inc.) and antibiotics.

### 5.3. Preparation of Enzyme-Assisted Hydrolysates of M. edulis

Domestically cultivated *M. edulis* specimens were initially washed with running tap water to remove any adherent materials. After washing, the shells were carefully removed, leaving only the muscle tissue. The isolated mussel muscles were then subjected to freeze-drying using a vacuum freeze dryer (SAMWON Freezing engineering Co., Bucheon, Republic of Korea). The freeze-dried mussel muscles were further processed using a grinder, which mechanically reduced them into a fine powder (SHINIL Electronics Co., Ltd., Incheon, Republic of Korea). The resulting *M. edulis* was homogenized by thorough mixing in distilled water to obtain a 10% solution. The pH of the solution was adjusted to 8.00, followed by addition of 1% enzyme (Protamex, Alcalase, and Flavourzyme). The enzymatic hydrolysis was performed for 24 h in a shaking incubator (SFDSM06, Jeio Tech, Seoul, Republic of Korea) at 120 rpm and 55 °C. Thereafter, the enzymes were inactivated by heating the reaction mixture at 100 °C for 10 min in a hot water bath (1D027013, Jeio Tech, Republic of Korea). After subsequent cooling, the solution was vacuum-filtered using a Buchner funnel with Whatman number 4 filter paper (Cat.#: 1004110; GE Healthcare, Amersham HP7 9NA, UK). The collected filtrate was freeze-dried and stored at −4 °C for further analysis. The control extract was prepared using a water extract that was not subjected to protease digestion (DW). Figure 1 shows a flow chart for the preparation of EMHs.

### 5.4. Proximate Composition Analysis

The proximate chemical composition of each hydrolysate sample was determined according to the Association of Official Analytical Chemists (AOAC) methods.

#### 5.4.1. Measurement of Hydrolysate Yield

The hydrolysate yield was measured gravimetrically using aluminum weighing dishes. Briefly, clean, dry dishes were preheated overnight at 60 °C in an oven to ensure the dishes were free of impurities and moisture. After subsequent cooling, the weight of the empty dishes was recorded, and 1 mL of hydrolysate was carefully transferred into them avoiding loss or contamination. The weighing dishes with hydrolysate were heated in an oven (500 °C) for 24 h to remove the moisture. Thereafter, the weight of the cooled dishes was taken, and hydrolysate yield was calculated as a percentage by dividing the difference in weight by the initial weight of the sample (1 mL = 1 g) and multiplying by 100.

#### 5.4.2. Measurement of Protein Content

The protein content was determined using the Kjeldahl method (Kjeltec^TM^ 8100, FOSS, Hillerød, Denmark). Briefly, the initial weight of a representative sample containing the protein was measured. Thereafter, the sample was digested with concentrated sulfuric acid (H_2_SO_4_) by adding a Kjeldahl catalyst tablet. The mixture was cooled and distilled using steam distillation. The process separated ammonia from other components of the mixture. The distilled ammonia was titrated using hydrochloric acid (HCl), and 1% boric acid was used as an indicator. The amount of acid required to neutralize ammonia was measured and used for calculating the nitrogen content in the original sample. Because proteins contain approximately 16% nitrogen, the protein content of a sample can be calculated using a conversion factor. By multiplying the measured nitrogen content by the appropriate factor (6.25), the protein content of the original sample was determined.

#### 5.4.3. Measurement of Lipid Content

The lipid content was determined gravimetrically using Soxhlet extraction (Fat Extractor E-500, Büchi Labortechnik AG, Flawil, Switzerland) with petroleum ether (DAEJUNG, Siheung-si, Gyeonggi-do, Republic of Korea). The initial weight of the homogenized samples was measured. The collection flasks were preheated overnight at 500 °C in an oven, and their empty weights were noted after cooling. The Soxhlet apparatus was set up, and the samples were placed inside a porous thimble that was fitted into the Soxhlet extractor. The extraction was carried out by adding a sufficient volume of petroleum ether. Thereafter, the lipid-containing collection flasks were heated to evaporate the remaining solvent. The weight of the cooled-down collection flasks was taken, and lipid content was calculated as a percentage by dividing the difference in weight with the initial weight of the sample and multiplying it by 100.

#### 5.4.4. Measurement of the Ash Content

The ash content was measured by incinerating the samples to ash in an electric muffle furnace (JSMF-30T, JSR Research Inc., Gongju City, Chungchungnam-Do, Republic of Korea) at 550 °C for 6 h [20]. Briefly, the samples were ground or homogenized to ensure uniformity. Empty porcelain crucibles were preheated in the electric furnace at a high temperature (often 500–600 °C) until their weight was constant. After cooling down the crucibles, their empty weight was taken, and the weighed samples were carefully placed in them. The crucible containing the sample was placed in the electric furnace preheated at 550 °C and combusted for 6 h. After complete ashing, the crucibles were carefully removed from the furnace using a pair of crucible tongs and allowed to cool to room temperature in a desiccator. The crucibles were then weighed on an analytical balance to determine the weight of the residual ash. The ash content was calculated by subtracting the weight of the empty crucible from the weight of the crucible with the residual ash. The weight difference represented the ash content of the original sample.

### 5.5. Evaluation of Cytotoxicity

The cytotoxicity of EMHs was measured using the MTT (3-(4,5)-dimethylthiazol−2-y1)-2, 5-diphenyltetrazolium bromide) assay [56]. C2C12 cells were seeded in a 96-well plate at 1 × 10^5^ cells/well and incubated at 37 °C for 24 h in the presence of 5% CO_2_. The cells were then treated with various concentrations of extracts (10–300 µg/mL) for 24 h. Following exposure to EMHs for 24 h, 20 µL of 5 mg/mL MTT reagent in phosphate-buffered saline (PBS) was added to each well and incubated at 37 °C for 1 h in the presence of 5% CO_2_. Thereafter, the solution was removed from all the wells, and 100 μL DMSO was added to solubilize the formazan formed. The absorbance in each well was measured at 540 nm using a microplate reader (Synergy^TM^ HT, Agilent Technologies, Santa Clara, CA, USA).

### 5.6. Evaluation of Cell Proliferation

The effect of distilled water (DW) and Protamex (PMH)-, Alcalase (AMH)-, and Flavourzyme (FMH)-assisted *M. edulis* hydrolysates on the proliferation of cells was evaluated using bromodeoxyuridine (BrdU) labeling and a detection kit (11647229001, Roche^®^ Life Science, Penzberg, Upper Bavaria, Germany) according to the vendor’s instructions [57].

### 5.7. Immunofluorescence Staining

Differentiated myotubes were washed with cold PBS, fixed, and permeabilized with chilled methanol followed by washing with phosphate-buffered saline with Tween-20 (PBST) three times. Fixed and permeabilized cells were blocked with 1% bovine serum albumin (BSA) containing 22.52 mg/mL glycine for 2 h and incubated with an MyHC-specific primary antibody (1:250 in 1% BSA) for 2 h at room temperature (25 °C). The cells were then incubated with goat antimouse IgG H&L (Alexa Fluor ^TM^ Plus 555; A32727)-conjugated secondary antibodies. Cell nuclei were stained with 300 nM 4′,6-diamidino-2-phenylindole (DAPI; Sigma-Aldrich). Fluorescence images of stained myotubes were visualized using a confocal microscope (Carl Zeiss, Oberkochen, Germany). The total number of nuclei, myotube coverage, and myotube diameter of the images were analyzed using the Myotube Analyzer software (Version 1.0.1) [58].

### 5.8. Western Blot Analysis

Whole-cell lysate was prepared using RIPA buffer (89901, Thermo Fisher Scientific, Waltham, MA, USA) containing a protease inhibitor cocktail and phosphatase inhibitor cocktail (GenDEPOT, Katy, TX, USA). Protein concentration was measured using the Pierce BCA Protein Assay Kit (23227, Thermo Fisher Scientific). Equal amounts of protein were mixed with the sample buffer and heated at 100 °C for 5 min. The proteins were electrophoresed on a 7.5% SDS polyacrylamide gel and transferred onto a nitrocellulose membrane (NC; Protran^TM^ 0.45 µm NC, Amersham, IL, USA). The membrane was blocked with 5% skim milk in Tween 20/Tris-buffered saline (TBST) at room temperature for 2 h. The membrane was then incubated with the primary antibody (1:1000 dilution) diluted in 5% BSA at 4 °C for 16 h. The blots were washed three times with TBST and incubated with secondary antibodies for 2 h (1:10,000) at room temperature. After washing three times with TBST, the protein bands were detected using an enhanced chemiluminescent substrate (Amersham) and quantified using the ImageJ software 1.51o (National Institute of Health, Bethesda, MD, USA).

### 5.9. Zebrafish Experiments

Adult zebrafish were purchased from a commercial supplier (World Fish Aquarium Corp, Jeju-do, Republic of Korea) and reared in clear acrylic tanks with specific conditions, including temperature of 28.5 ± 1 °C, 14/10 h light/dark cycle, and twice-daily feedings from Tetra GmgH (D-49304 Melle, Made in Germany). To establish the Dex-induced muscle atrophy model in zebrafish, individuals with a consistent body weight (3.25 ± 0.05 g) were divided into various groups: (1) control, (2) blank, (3) AMH (3%), and (4) maca (3%). All groups were provided with a recommended zebrafish-exclusive diet containing the specified samples, except for the blank and control groups. As zebrafish can absorb drug compounds through their skin and gills, formulated dexamethasone solution of 0.01% concentration was used to treat the fish for 1 h per day prior to feeding. The Dex exposure lasted for 10 days, and the samples containing diet were introduced three days before Dex exposure. To assess muscle atrophy in zebrafish, their swimming-related behaviors, including total distance moved, velocity, mobility state, and immobility state, were monitored and analyzed using Ethovision XT software 15.0 (Noldus Information Technology, Wageningen, The Netherlands).

### 5.10. Determination of Amino Acid Composition of AMH

The amino acid composition of the selected sample (AMH) was performed using modified Rutherfurd’s method [59]. Briefly, 5 mg of AMH was dissolved in 1 mL of distilled water (DW) and hydrolyzed using HCl for 24 h at 110 °C followed by desiccation. Cystine hydrolysis was performed by HCl hydrolysis after peroxidation, and tryptophan hydrolysis was performed by hydrolysis with methanesulfonic acid. Following hydrolysis, the amino acids were derivatized by mixing them with phenylisothiocyanate (PITC) in a derivatization solution. This solution, consisting of MeOH, H_2_O, TEA, and PITC, was mixed for 30 min at room temperature. The resulting derivatized samples were then dissolved in solvent A, composed of NaHAc, TEA, EDTA, and CH_3_CN, at pH 6.1. After the reagent from the derivatization samples was dried, corrected supernatants were injected into an HPLC system (Agilent 1260 Series) equipped with a specific column (Waters Nova-Pak C18–4 μm; 3.9 × 300 mm). The solvent B conditions were set at 60% CH_3_CN and 0.015% EDTA, and detection was performed at 254 nm.

### 5.11. Statistical Analysis

Results are expressed as means ± standard deviations (SDs). Group results were analyzed using the *t*-test or one-way ANOVA using GraphPad PRISM 6 (GraphPad Software, San Diego, CA, USA). Significance compared with the values for the control is denoted as ^#^ *p* < 0.05, ^##^ *p* < 0.01, ^###^ *p* < 0.001, and ^####^ *p* < 0.0001, and significance compared with the Dex group is denoted as * *p* < 0.05, ** *p* < 0.01, *** *p* < 0.001, and **** *p* < 0.0001.

## Figures and Tables

**Figure 1 marinedrugs-21-00623-f001:**
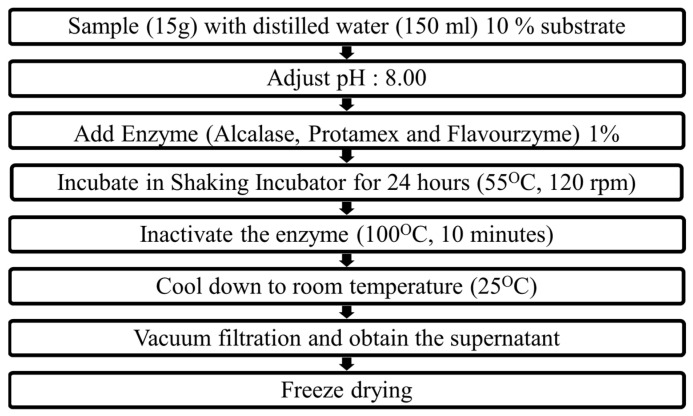
Enzyme-assisted *M. edulis* hydrolysate preparation process.

**Figure 2 marinedrugs-21-00623-f002:**
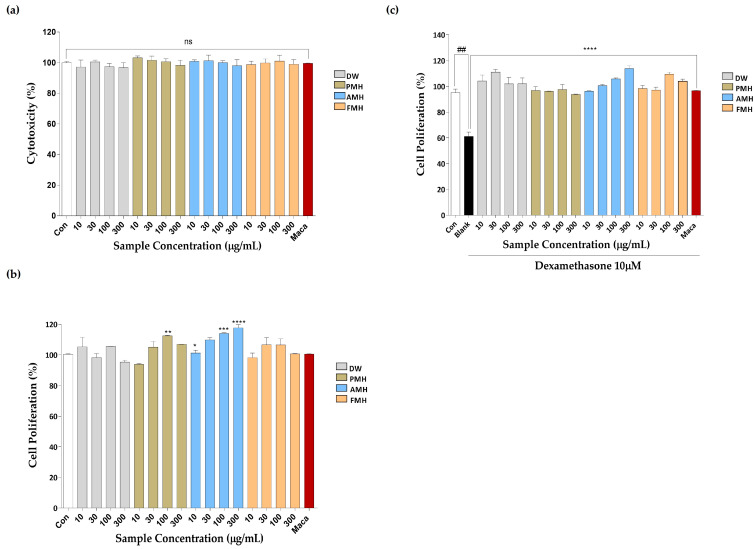
Effect of DW, PMH, AMH, and FMH on C2C12 cell proliferation. (**a**) Cytotoxicity, (**b**) cell proliferation, and (**c**) Dex-induced C2C12 cell proliferation. Experiments were carried out in triplicate, and the results are presented as means ± SD (*n* = 3). ^##^ *p* < 0.01 vs. control; * *p* < 0.05, ** *p* < 0.01, *** *p* < 0.001, **** *p* < 0.0001 vs. blank and “ns”: not significant.

**Figure 3 marinedrugs-21-00623-f003:**
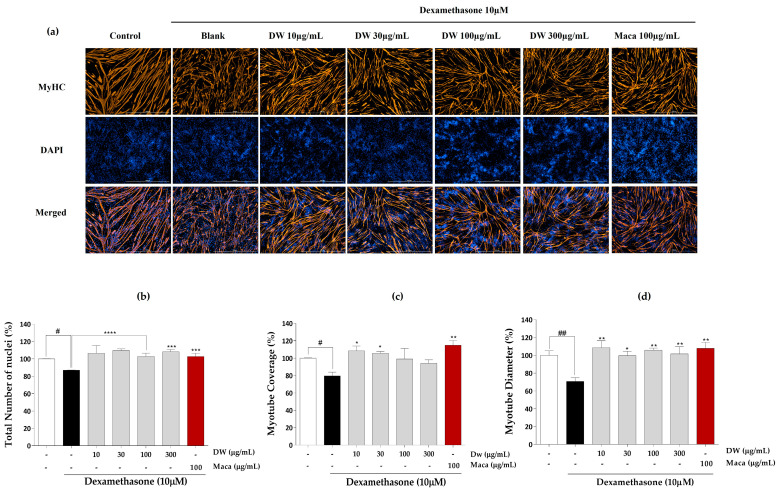
Immunofluorescence images of DW. (**a**) Immunostaining images with MyHC, (**b**) total number of nuclei, (**c**) myotube coverage, and (**d**) myotube diameter. Experiments were carried out in triplicate, and the results are presented as means ± SD (*n* = 3). ^#^ *p* < 0.05 and ^##^
*p* < 0.01 vs. control; * *p* < 0.05, ** *p* < 0.01, *** *p* < 0.001, and **** *p* < 0.0001 vs. blank.

**Figure 4 marinedrugs-21-00623-f004:**
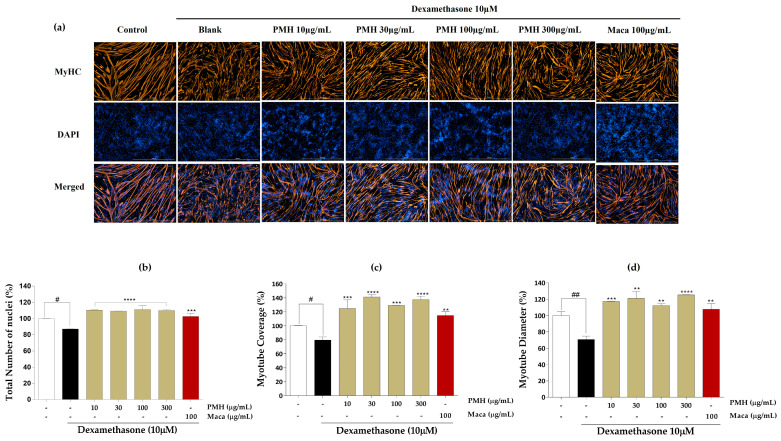
Immunofluorescence images of PMH. (**a**) Immunostaining images with MyHC, (**b**) total number of nuclei, (**c**) myotube coverage, and (**d**) myotube diameter. Experiments were carried out in triplicate, and the results are presented as means ± SD (*n* = 3). ^#^ *p* < 0.05 and ^##^ *p* < 0.01 vs. control; ** *p* < 0.01, *** *p* < 0.001, and **** *p* < 0.0001 vs. blank.

**Figure 5 marinedrugs-21-00623-f005:**
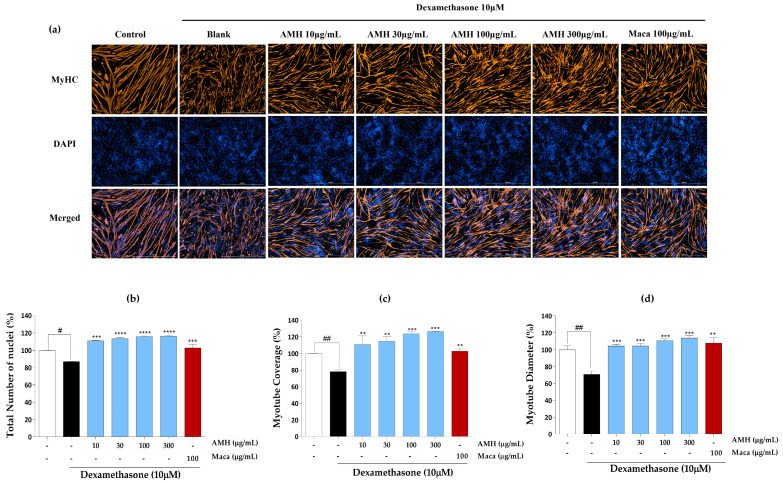
Immunofluorescence images of AMH. (**a**) Immunostaining images with MyHC, (**b**) total number of nuclei, (**c**) myotube coverage, and (**d**) myotube diameter. Experiments were carried out in triplicate, and the results are presented as means ± SD (*n* = 3). ^#^ *p* < 0.05 and ^##^ *p* < 0.01 vs. control; ** *p* < 0.01, *** *p* < 0.001, and **** *p* < 0.0001 vs. blank.

**Figure 6 marinedrugs-21-00623-f006:**
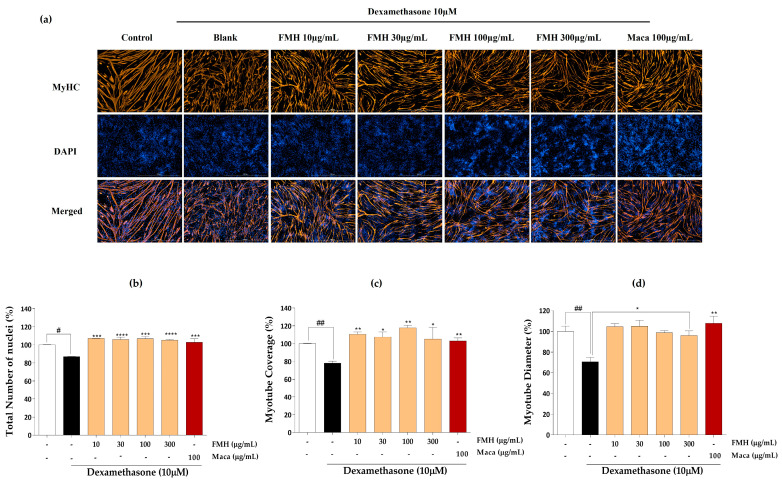
Immunofluorescence images of FMH. (**a**) Immunostaining images with MyHC, (**b**) total number of nuclei, (**c**) myotube coverage, and (**d**) myotube diameter. Experiments were carried out in triplicate, and the results are presented as means ± SD (*n* = 3). ^#^ *p* < 0.05 and ^##^ *p* < 0.01 vs. control; * *p* < 0.05, ** *p* < 0.01, *** *p* < 0.001, and **** *p* < 0.0001 vs. blank.

**Figure 7 marinedrugs-21-00623-f007:**
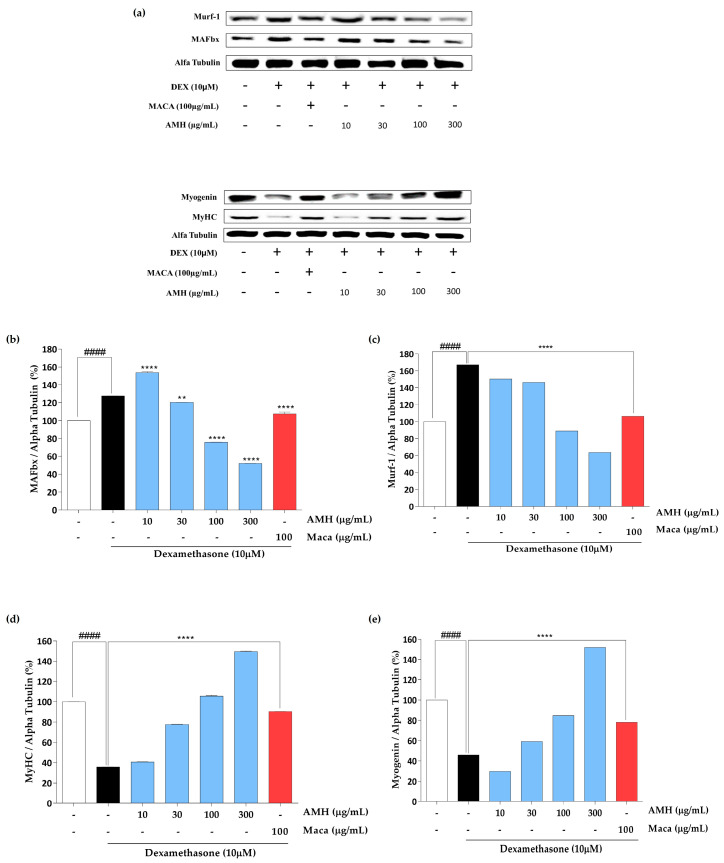
Assessment of AMH on proteins responsible for muscle atrophy. (**a**) Protein expressions evaluated by Western blot; (**b**) quantification of the expression of MAFbx protein, (**c**) Murf-1, (**d**) MyHC, and (**e**) myogenin. The data are represented as mean ± SE. ^####^ *p* < 0.0001 vs. control; dexamethasone group ** *p* < 0.01, and **** *p* < 0.0001 vs. blank.

**Figure 8 marinedrugs-21-00623-f008:**
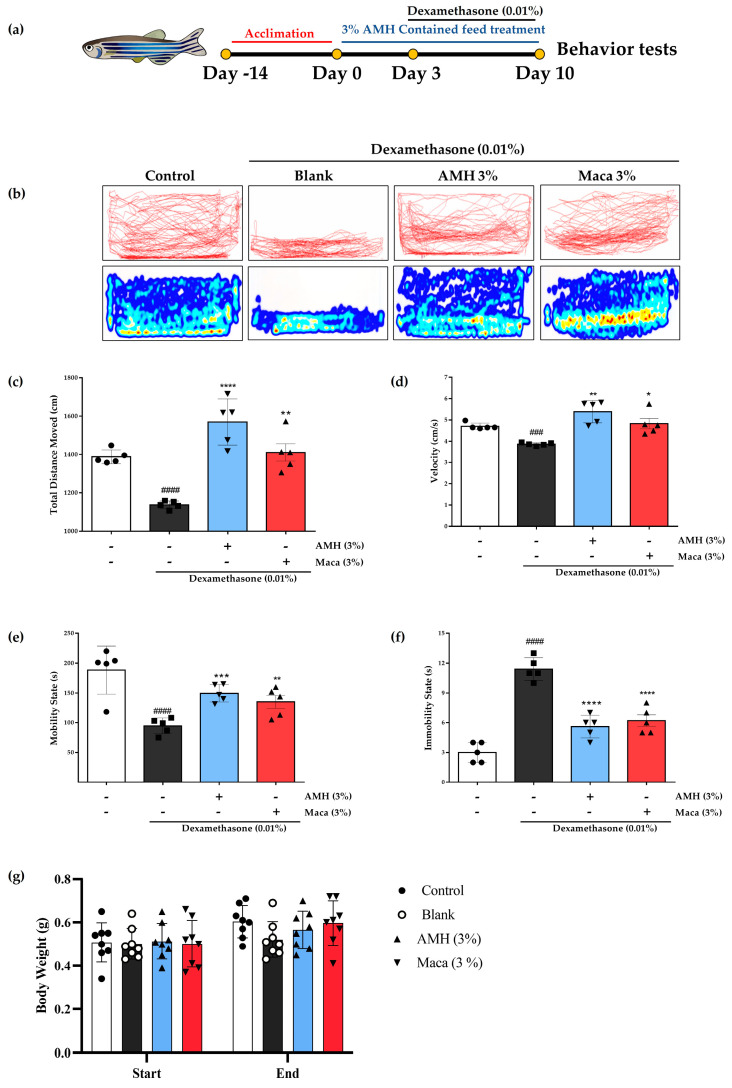
Recovery of Dex-induced muscle atrophy in zebrafish by AMH administration. (**a**) Zebrafish experimental procedure and (**b**) locomotion tracking of Dex-induced muscle atrophy zebrafish model. The graphically quantified values of (**c**) total distance moved, (**d**) velocity, (**e**) mobility state, and (**f**) immobility state. (**g**) Graphical illustration of initial and final body weight change of the experimented zebrafish. The data are represented as mean ± SE. ^###^ *p* < 0.001, ^####^ *p* < 0.0001 vs. control; * *p* < 0.05, ** *p* < 0.01, *** *p* < 0.001, and **** *p* < 0.0001 vs. blank.

**Table 1 marinedrugs-21-00623-t001:** Hydrolysate yield and proximate analysis of EMHs.

Sample	Hydrolysate Yield (%)	Proximate Composition (%)			
		Protein	Lipid	Polysaccharide	Ash
DW	44.40 ± 1.60	43.61 ± 1.36	5.86 ± 0.56	30.20 ± 1.38	20.32 ± 0.15
PMH	72.90 ± 2.90 ***	56.18 ± 0.77 **	6.48 ± 0.18	21.44 ± 1.27 **	15.90 ± 0.03 ***
AMH	76.37 ± 0.77 ***	57.33 ± 2.40 **	6.62 ± 0.03	21.61 ± 0.53 **	14.70 ± 0.28 ***
FMH	63.22 ± 2.02 ***	51.95 ± 0.75 *	6.19 ± 0.17	23.51 ± 0.80 **	18.51 ± 0.09 ***

The experiments were triplicated, and data are represented as average ± SD. Significance is denoted compared to DW. * *p* < 0.05, ** *p* < 0.01, *** *p* < 0.001.

**Table 2 marinedrugs-21-00623-t002:** Amino acid profile of AMH.

Amino Acid (AA)	mg/100 g
Arginine (ARG)	3062
Histidine (HIS)	824
Isoleucine (ISO)	1977
Leucine (LEU)	3016
Lysine (LYS)	3718
Methionine (MET)	896
Phenylalanine (PHE)	1694
Threonine (THR)	231
Tryptophan (TRY)	394
Valine (VAL)	2276
Alanine (ALA)	2668
Cystine (CYS)	32
Glutamic acid (GLA)	5883
Proline (PRO)	1918
Serine (SER)	2482
Tyrosine (TYR)	1332
Glycine (GLY)	3732
Aspartic acid (ASA)	4656
Total	43,158
EAA	20,167 (46.73% of Total AAs)
Non-EAA	22,991 (53.27% of Total AAs)
NCAA	10,539 (24.42% of Total AAs)
AAA	342 (7.92% of Total AAs)
BCAA	7269 (16.84% of Total AAs)

EAA: essential amino acids (ARG, HIS, ISO, LEU, LYS, MET, PHE, THR, TRY, and VAL), Non-EAA: non-essential amino acids (ALA, CYS, GLA, PRO, SER, TYR, GLY, and ASA), NCAA: negatively charged amino acids (ASA and GLA), AAA: aromatic amino acids (PHE, TRY, and TYR), BCAA: branched-chain amino acids (VAL, LEU, and ISO).

## Data Availability

Data are contained within the article and Appendix A.

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
