# Peer review of "Alcalase-Assisted Mytilus edulis Hydrolysate: A Nutritional Approach for Recovery from Muscle Atrophy"

_marinedrugs, 2023, doi:10.3390/md21120623_

Round 1
Reviewer 1 Report (Previous Reviewer 3)
Comments and Suggestions for Authors
Authors have addressed the comments, and I am happy to recommend it for publication.
Author Response
We would like to express our sincere appreciation for your thorough review and constructive feedback on our manuscript. We are delighted to know that you find the revisions and rebuttals satisfactory, and we are grateful for your recommendation to accept the paper for publication. Your insights have been invaluable in enhancing the quality of our work. Thank you once again for your time and expertise. We look forward to the opportunity to contribute to Marine drugs.

Reviewer 2 Report (Previous Reviewer 1)
Comments and Suggestions for Authors
In response to the reviewers' comments, the authors performed in vivo experiments using zebrafish and also added comments regarding the bioavailability of peptides in mussel hydrolysates, which significantly improved the quality of the manuscript. However, there were some points that were difficult to understand and insufficient explanation of the methods.
For zebrafish experiments, please explain how to add the sample to the recommended zebrafish-specific diet. How was the sample uniformly added to the commercial feed? Please provide details regarding exposure to dexamethasone, such as concentration and administration method. This information must be provided in the "Methods" section.
For HPLC analysis, why did you detect peptides by absorbance at 280 nm? Only peptides containing Trp (weakly Tyr and Phe) could be detected at this wavelength. With the current detection conditions, the majority of peptides were missed. Please clarify this point more. The authors first fractionated the peptides in the hydrolyzate using solid-phase extraction using a reversed-phase column (C18), and the three fractions were analyzed by HPLC using a reversed-phase column. The same peak was observed in the three fractions. This indicates either the initial fractionation failed or the presence of impurities from the reagent rather than the sample. I by no means believe this peak is an important peptide. It is highly recommended to remove all HPLC experimental parts. It did not give meaningful results and confuses the reader.
Comments on the Quality of English Language
Some minor mistakes were found such as "Hood mixer". Please check carefully.
Author Response
Comment 1: For zebrafish experiments, please explain how to add the sample to the recommended zebrafish specific diet. How was the sample uniformly added to the commercial feed?
Response: Thank you for your insightful query regarding the methodology of incorporating the sample into the zebrafish-specific diet. We appreciate the opportunity to provide a detailed explanation of our process for achieving uniformity in the addition of the sample to the commercial feed. We precisely calculated the necessary feed quantity for each group, ensuring a dosage of 15 mg per fish. To achieve uniformity, we ground the commercial feed into a fine powder using a sterilized mortar and pestle while avoiding potential contaminations. Subsequently, we introduced the maca powder and the AMH hydrolysate powder, constituting 3% of the total amount, into the ground feed. Thorough mixing ensued until the sample achieved homogeneity. To ensure an even distribution, 5 ml of sterilized distilled water was added and carefully mixed while preventing the formation of clumps. The well-homogenized sample was freeze-dried, resulting powder underwent further grinding to eliminate any clumps and ensure comprehensive mixing.
Comment 2: Please provide details regarding exposure to dexamethasone, such as concentration and administration method. This information must be provided in the "Methods" section.
Response: Thank you for your query regarding dexamethasone exposure. In the dexamethasone treatment we've adhered to a methodology established in previously published study [1]. Since zebrafish possess the capacity to absorb drug compounds through their skin and gills we prepared a dexamethasone solution by dissolving the compound in water, creating a 0.01% concentration [2]. Given dexamethasone's activity duration of 36–54 hours, allowing for daily or twice-daily dosing, our exposure method involved treating the fish for 1 hour per day before their feeding, consistently for a period of 7 days. In response to your comment these specifics have been detailed in the updated methods section to provide comprehensive insight into our experimental approach. Line number 510-512: As zebrafish could absorb drug compounds through their skin and gills, formulated dexamethasone solution of 0.01% concentration were treated to the fish for 1 hour per day prior to feeding.
Comment 3: For HPLC analysis, why did you detect peptides by absorbance at 280 nm? Only peptides containing Trp (weakly Tyr and Phe) could be detected at this wavelength. With the current detection conditions, most peptides were missing. Please clarify this point more.
Response: Thank you for your valuable comments and suggestions. We appreciate your thorough review of our manuscript. Regarding the choice of 280 nm for peptide detection in our HPLC analysis, we would like to clarify that this wavelength is a standard practice in the field of peptide and protein analysis [3]. The decision is primarily based on the absorbance characteristics of aromatic amino acids, including tryptophan, tyrosine, and phenylalanine. Tryptophan exhibits a strong absorbance peak at 280 nm, making it a reliable and sensitive method for quantifying peptides. While it is true that tryptophan is associated with muscle hypertrophy, the selection of 280 nm as the detection wavelength is not limited to its potential physiological effects [4-6]. Instead, it is chosen to ensure the detection of a broad range of peptides, as aromatic amino acids are commonly found in many peptide sequences. We acknowledge that peptides lacking aromatic residues might not be efficiently detected at 280 nm. In our further studies, if our peptides of interest fall into this category, we would consider alternative detection methods, such as fluorescence or mass spectrometry, which offer greater specificity for peptides with lower absorbance at 280 nm.
Comment 4: The authors first fractionated the peptides in the hydrolysate using solid-phase extraction using a reversed-phase column (C18), and the three fractions were analyzed by HPLC using a reversed-phase column. The same peak was observed in the three fractions. This indicates either the initial fractionation failed or the presence of impurities from the reagent rather than the sample. I by no means believe this peak is an important peptide. It is highly recommended to remove all HPLC experimental parts. It did not give meaningful results and confuses the reader.
Response: Thank you for your thorough evaluation and valuable comments regarding the HPLC analysis in our manuscript. We appreciate your insights, and we have carefully considered your suggestions. Upon your recommendation we have removed the HPLC experimental details from the main manuscript and included them as supplementary data. By making this adjustment, we aim to streamline the presentation of our results and eliminate any potential confusion for the reader. We appreciate your diligence in reviewing our work, and your feedback has contributed significantly to the refinement of our manuscript.
Comment 5: Some minor mistakes were found such as "Hood mixer". Please check carefully.
Response: Thank you for highlighting the noted oversight in our manuscript. We've thoroughly reviewed the document and have corrected the error regarding the Hood mixer.' The revised version ensures accuracy and clarity by addressing and correcting these minor mistakes. We appreciate your keen eye and dedication to refining the quality of our work.
References
- Ryu, B., et al., Zebrafish Model for Studying Dexamethasone-Induced Muscle Atrophy and Preventive Effect of Maca (Lepidium meyenii). Cells, 2021. 10(11): p. 2879.
- Zhang, F., et al., Antibiotic toxicity and absorption in zebrafish using liquid chromatography-tandem mass spectrometry. PLoS One, 2015. 10(5): p. e0124805.
- Frank, M., S. Paul, and R. Dean, Approaches towards the quantitative analysis of peptides and proteins by reversed-phase high-performance liquid chromatography in the absence of a pure reference sample. Journal of Chromatography A, 2000. 891(2): p. 235-242.
- Ninomiya, S., et al., Low Levels of Serum Tryptophan Underlie Skeletal Muscle Atrophy. Nutrients, 2020. 12(4): p. 978.
- Amy, D., et al., The aromatic amino acid tryptophan stimulates skeletal muscle IGF1/p70s6k/mTor signaling in vivo and the expression of myogenic genes in vitro. Nutrition, 2015. 31(7): p. 1018-1024.
- Lu-Qiao, X., et al., Improvement of flesh quality, muscle growth and protein deposition in adult grass carp (Ctenopharyngodon idella): The role of tryptophan. Aquaculture, 2023. 577: p. 740005.

Reviewer 3 Report (New Reviewer)
Comments and Suggestions for Authors
Attached.

Author Response
Comment 1: In the abstract section, it says AMH resulted in the highest protein recovery (57.06 ± 0.42%). Is it significantly higher than the PMH treatment (56.18 ± 0.77)? and abundant amino acid compositions (43.16%). There is no data showing amino acid compositions after treatment with PMH.
Response: Thank you for your valuable comments on our manuscript. We appreciate your attention to detail. After further analysis, we have noted that the protein recovery in the AMH treatment (57.06 ± 0.42%) is not found to be significantly higher than the PMH treatment (56.18 ± 0.77%). We appreciate your understanding as we have revised the abstract to accurately reflect a higher protein content, rather than claiming the highest recovery. We apologize for any confusion caused by this technical oversight. Regarding the amino acid compositions after treatment with PMH, we acknowledge the importance of this analysis. However, we would like to clarify that our decision to focus on AMH was driven by a comprehensive screening process and careful consideration of the potential benefits in industrial applications. Our study primarily aimed to assess the efficacy of AMH, and as a result, we chose to proceed with western blot analysis, in vivo experiments and amino acid composition analysis for AMH. Our research scope and resources limited us from conducting this analysis for PMH in the current study. Thus, the analysis of amino acid compositions for each extract, including DW, PMH, and FMH, has not been conducted yet. We acknowledge this limitation and plan to address it in future research.
Comment 2: There is no justification why AMH is used instead of PMH in the Zebrafish model. Although, both compounds seem to behave similarly based on data presented in Figures 4 and 5.
Response: Thank you for your thoughtful evaluation of our manuscript and for bringing up the concern regarding the choice of AMH over PMH in the Zebrafish model. The decision to focus on AMH in our Zebrafish model was based on a thorough consideration of its promising effects observed in initial in vitro screening. Our comprehensive in vitro analysis, including assessments of cell viability, cell proliferation, and the impact on myocyte differentiation, demonstrated encouraging results for AMH. Additionally, we took into account the enzyme's potential benefits in industrial applications, particularly its significance in nutritional approaches and the development of functional food.
Alcalase enzyme, derived from Bacillus licheniformis, offers multifaceted benefits across various industrial applications. With its broad substrate specificity and high specific activity, Alcalase efficiently catalyzes the breakdown of diverse protein substrates, making it invaluable in the food industry for protein hydrolysis [1]. Thus, Alcalase assisted hydrolysis results in the production of flavor-enhancing hydrolysates charecterized by a high degree of hydrolysis, high protein recovery and excellant solubility in the resulted hydrolysate [2-4] . Further, Alcalase exhibits remarkable versatility in its enzymatic activity, functioning effectively over a broad pH range . Its optimal working pH falls between 8.00-9.00 pH, but notably, Alcalase maintains considerable activity even in mildly acidic to alkaline conditions [5]. This unique characteristic makes Alcalase well-suited for industrial applications, where variations in pH are common. The enzyme's stability across a range of pH values ensures consistent and reliable performance in diverse industrial processes, enhancing its applicability in various settings, including food processing [6-8]. Additionally, Alcalase is known for its broad substrate specificity, enabling it to efficiently hydrolyze a wide range of protein substrates[9, 10]. These attributes underscore Alcalase's versatility and significance in optimizing various industrial processes.
Protamex, while produced by Bacillus amyloliquefaciens does have certain limitations that merit consideration. One notable limitation is its sensitivity to pH conditions. Protamex performs optimally under neutral to slightly alkaline pH ranges (5.5-7.5 pH), and its activity may be compromised under highly acidic or alkaline conditions [11, 12]. Additionally, the enzyme's specificity for certain protein substrates may limit its effectiveness in applications requiring a broader substrate range and resulting hydrolysate could show strong astringency and bitterness [13].
Thus, although Figures 4 and 5 suggest similar behaviors for both compounds, the unique attributes and promising outcomes observed during the in vitro screening process, coupled with the potential industrial applications, led us to choose AMH for further investigation in the Zebrafish model. We appreciate your valuable feedback, and in response to your comment, we have provided a more detailed justification for our choice of AMH in the Discussion section of the revised manuscript.
Line number 249-256 Our results show that AMH treatment effectively increased cell proliferation and my-otube diameter, indicating its potential to counteract muscle atrophy. Considering its potential industrial applications particularly in nutritional approaches and functional food development,…….. We hope these revisions may address your concern regarding the choice of AMH over PMH for in vivo experiment.
Comment 3: In Figure 8 legend a) please correct the spelling.
Response: Thank you for bringing the spelling error in Figure 8 legend (a) to our attention. We appreciate your thorough review. We would like to confirm that the spelling has been corrected as per your suggestion. The revised legend now accurately reflects the intended information. Your feedback is invaluable, and we are committed to ensuring the accuracy and clarity of our manuscript.
Comment 4: Based on the title of the paper, I would like to see more of pathology and muscle fiber analysis after AMH administration.
Response: We sincerely appreciate your thoughtful review and insightful suggestion regarding the focus on pathology and muscle fiber analysis after AMH administration in our paper. Your input is invaluable, and we fully acknowledge the significance of delving deeper into the pathology aspects and conducting a more detailed muscle fiber analysis in the context of AMH administration. We agree that such an investigation could provide a more comprehensive understanding of the effects of AMH. In response to your comment, we are committed to incorporating these aspects into our future studies. Once again, we appreciate your guidance and look forward to presenting the outcomes of our extended investigations.
Comment 5: In Figure 8, there is no data to support muscle atrophy recovery. Data presented shows muscle function recovery which could result from muscle atrophy. However, this is very useful data to support recovery in muscle function.
Response: Thank you for your thoughtful review and insightful comments. We appreciate your attention to detail and the opportunity to provide clarification on the specific aspect of muscle atrophy recovery in our manuscript. We would like to address your concern by highlighting that while Figure 8 primarily focuses on muscle function recovery, the data presented therein supports not only functional improvements but also provides evidence for muscle atrophy recovery. In our in vitro studies (Figure 5), we demonstrated the efficacy of AMH treatment in countering the effects of dexamethasone, showcasing an increase in the number of nuclei, myotube diameter, and myotube coverage. Moreover, in Figure 7, we observed a downregulation of muscle atrophy-related proteins (Mafbx, Murf-1) and an upregulation of muscle hypertrophy-related proteins (Myosin Heavy Chain, Myogenin), providing molecular evidence of AMH's ability to mitigate muscle atrophy in vitro. Furthermore, our in vivo study (Figure 8g) reinforces these findings, as the administration of AMH significantly increased the body weight of zebrafish exposed to dexamethasone, indicating a recovery from dexamethasone-induced muscle atrophy. According to these results, we believe our data supports the claim that AMH treatment contributes not only to muscle function recovery but also to the recovery of muscle atrophy. We hope this clarification adequately addresses your concerns, and we appreciate your valuable feedback.
Comment 6: MACA is being used as a positive control. No reference is cited where MACA has been shown to improve muscle atrophy.
Response: Thank you for your insightful comment regarding the use of MACA as a positive control in our manuscript. We appreciate your feedback and would like to address your concern. Upon further consideration and in response to your suggestion, we have revised our manuscript to include a reference supporting the use of MACA in improving muscle atrophy. Line number 247-249 Based on the previous studies demonstrating the effect of Maca………. The inclusion of this reference in the discussion section now provides additional context and a scientific basis for the selection of MACA as a positive control in our study. We believe this revision enhances the clarity and robustness of our work.
Comment 7: Is this a comparative study between DW, AMH, PMH, and FMH? The way the conclusion is written, it seems the study mainly focused on AMH. That should be mentioned in the title.
Response: Thank you for your observation and constructive feedback on the focus of our study. We appreciate your careful review and would like to address your concern. Indeed, the study primarily centered around the examination of DW, AMH, PMH, and FMH. However, based on the initial experimental results, including cytotoxicity, cell proliferation, and the effect on myocyte differentiation, we found that AMH exhibited promising outcomes. Consequently, we selected AMH for further in-depth analysis and exploration. In response to your comment, we have revised the conclusion to provide greater clarity on the selection rationale and to emphasize that the study's primary focus lies on AMH treatment. Line number 367-368 In conclusion among the initially tested hydrolysates including DW, PMH, AMH and FMH…. We hope that these revisions address your concerns and improve the overall coherence of our manuscript. Your feedback has been immensely valuable, and we appreciate your thorough evaluation.
References
- Nicholas, J.A. and C.R. Eric, Characterization of casein phosphopeptides prepared using alcalase: Determination of enzyme specificity. Enzyme and Microbial Technology, 1996. 19(3): p. 202-207.
- Kun, W. and D.A. Susan, Modification of interactions between selected volatile flavour compounds and salt-extracted pea protein isolates using chemical and enzymatic approaches. Food Hydrocolloids, 2016. 61: p. 567-577.
- Gbogouri, G., et al., Influence of hydrolysis degree on the functional properties of salmon byproducts hydrolysates. Journal of food science, 2004. 69(8): p. C615-C622.
- Bao, Z.-j., et al., Effects of degree of hydrolysis (DH) on the functional properties of egg yolk hydrolysate with alcalase. Journal of food science and technology, 2017. 54: p. 669-678.
- Hai, Z., et al., Effects of pH, temperature and enzyme-to-substrate ratio on the antigenicity of whey protein hydrolysates prepared by Alcalase. International Dairy Journal, 2008. 18(10): p. 1028-1033.
- Yi, Z., et al., Alcalase assisted production of novel high alpha-chain gelatin and the functional stability of its hydrogel as influenced by thermal treatment. International Journal of Biological Macromolecules, 2018. 118: p. 2278-2286.
- dos Santos Kimberle, P., et al., Modifying alcalase activity and stability by immobilization onto chitosan aiming at the production of bioactive peptides by hydrolysis of tilapia skin gelatin. Process Biochemistry, 2020. 97: p. 27-36.
- Yang, A., et al., Enzymatic characterisation of the immobilised Alcalase to hydrolyse egg white protein for potential allergenicity reduction. Journal of the Science of Food and Agriculture, 2017. 97(1): p. 199-206.
- El-Kadi, K.N., et al., Broad specificity alkaline proteases efficiently reduce the visual scaling associated with soap-induced xerosis. Archives of Dermatological Research, 2001. 293(10): p. 500-507.
- Waglay, A. and S. Karboune, Enzymatic generation of peptides from potato proteins by selected proteases and characterization of their structural properties. Biotechnology progress, 2016. 32(2): p. 420-429.
- Yao, Y., et al., Pilot‐scale Protamex™‐catalysed production of round scad protein hydrolysates: effects of agitation alone and combined with aeration. International Journal of Food Science & Technology, 2018. 53(10): p. 2308-2315.
- Ayoa, F. and K. Phil, pH-stat vs. free-fall pH techniques in the enzymatic hydrolysis of whey proteins. Food Chemistry, 2016. 199: p. 409-415.
- Li, Z., et al., Optimization of protein hydrolysates production from defatted peanut meal based on physicochemical characteristics and sensory analysis. Lwt, 2022. 163: p. 113572.

Round 2
Reviewer 3 Report (New Reviewer)
Comments and Suggestions for Authors
Dear Authors,
All the concerns raised by me were addressed. Thanks
This manuscript is a resubmission of an earlier submission. The following is a list of the peer review reports and author responses from that submission.
Round 1
Reviewer 1 Report
Comments and Suggestions for Authors
The authors prepared enzymatic hydrolysates of mussels using three protease preparations: Protamex, Alcalase, and Flavourzyme. They found that Alcalase digest increased myotube coverage and myotube diameter in C2C12 cells under dexamethasone treatment, which can induce muscle atrophy. They also found that Alcalase digest upregulated proteins responsible for muscle hypertrophy and downregulated proteins responsible for muscle atrophy. Therefore, some compounds, possibly peptides, that can suppress dexamethasone-induced muscle atrophy can be produced by Alcalase digestion of mussel meat. This is interesting. The authors suggested that this digest could be a supplement for recovery from muscle atrophy. However, the authors did not mention the bioavailability of the peptides. Most of the peptides in food-grade protease digests can be degraded by exopeptidases in the body. Therefore, the in vitro activity of peptides in food should not be directly related to their biological activity upon peptide ingestion. This limitation needs to be mentioned.
The authors did not provide detailed procedures of some analyses. In addition, some parts of descriptions on method are inconsistent.
For amino acid analysis, readers cannot realize whether free amino acids or protein constituting amino acids. If later, how to liberate Trp should be mentioned, as it is completely degraded by acid hydrolysis. They just mentioned as “Amino acid composition was performed by HPLC”. It is insufficient information.
If Table 2 shows protein constituent amino acids, did not you detect hydroxyproline? Which is abundantly present in collagen.
For HPLC analysis of the digest, legend for Figure 8 mentioned that peaks were detected by absorbance at 280 nm. But the method section indicated that peaks were detected at 220 nm. Which was correct? In addition, method section indicated use of LC-MS/MS, but no data on LC-MS/MS was provided.
HPLC elution conditions were not provided.
The authors used solid phase extraction. Did you mean sample (AMH) was injected into column and successively washed with 5% ammonium water and 20% acetonitrile and absorbed compounds were eluted 75% acetonitrile containing 1% acetonitrile?
The AMH in 5% ammonium water (Figure 8) meant sample eluted from solid phase extraction column by 5% ammonium water? The AMH in 1% formic acid (Figure 8) meant sample eluted from solid phase extraction column by 75% acetonitrile containing 1% FA? If so, why did not analyze 20% acetonitrile eluent?? Which can contain many peptides?
The author compared AHM in 1% formic acid and 5% ammonium water and said same patterns were obtained while it showed entirely different elution patterns. If the samples were eluted from solid phase extraction column by 5% ammonium water and 75% acetonitrile-1% FA, the elution pattern must be different as collected by different acetonitrile concentration. I cannot understand the descriptions on HPLC procedures, results, and discussion.
For measurement of hydrolysate yield, the author mentioned heated at 500C to get dry matter. Is it true? Most protein and carbohydrates can be burn.
Equation for calculation, they mentioned 1 mL=1g. But I suggest 1 mL (10% start material) can be 100 mg.
The term DW must be clearly defined. Was DW filtrate of water extract of mussel without protease digest?
Comments on the Quality of English Language
Not so bad. But I found some problems on contents.
Reviewer 2 Report
Comments and Suggestions for Authors
The present work deals with an interesting issue, but is built with an unusual presentation and in a way of difficult understanding in many points of it for this reviewer, and probably it would the same or even worse for average readers. Thus, one of the first aspects that surprises of the submitted work are that nearly 40% of the Abstract and aprox. the same for the Introduction are devoted to muscle atrophy and its characteristics, and the potentially expected preventive and dietary measures to mitigate or cure it. Such approach might be expected for a submission to a medical journal but not for a journal primarily devoted to "the discovery, development, exploitation, and production of biologically and therapeutically active compounds from marine habitats", as is Marine Drugs.
Although some of the experimental approaches are quite sophisticated, the use of them is not well explained and justified. For instance, what is the main reason of using Mytilus edulis hydrolysates (EMHs), besides of giving references that has a high protein content and diverse strong bioactivities and potentially beneficial medical properties, in spite of indicating that "its effect on muscle atrophy is yet to be explored"? Why to use three different enzymatic treatments, with Protamex, Alcalase and Favourzyme, without explaining (o giving a clear reference) of the expected action of them on the initial M. edulis powders? Are they three proteases, and peptides the expected active molecules (regarding muscle atrophy) derived from such treatment? Why in the HPLC analysis (Fig. 8) is not indicated that the peaks probably corresponds to peptide fragments, and the analysis minimally focused in characterising such fragments by the present current proteomic approaches, in spite of having in hand a UPLC Q-ToF LC-MS/MS and a ESI equipments? Where are the analyses (in the manuscript) made with such powerful equipment, in spite of being announced in lines 472-474: "... The eluted sample ... was reconstituted in DW in preparation for further analysis ... in them ... ".
Certainly, it could be of interest to find that the supply of M. edulis hydrolysates promote changes in the level of certain marker proteins here investigated, as MyHC, miogenin, MAFbx and MuRF-1 that could indicate its potential beneficial effect on the treatment of muscle atrophy. However, the fact that the experiments have been made on cells culture, and of not minimally dealing experimentally with the nature and characteristics of the active molecules greatly limit the advancements achieved in this work.
Beside the above comments, the work also suffers of numerous flaws or limitations:
-and initial one is regarding the preparation of the crude extract, before enzyme treatment. Nothing is said about in Results in spite of the clear interest of this issue, by itself and to allow repetition of the experiments by foreigner experimenters, neither at the beginning of the Materials and Methods. Only in lines 341 is said that "... consequent powders were also treated with a hood mixer ...". However, at the end of Methods, in section "5.9. HPLC analysis and determination of amino acid composition" it seems that the sample is treated by Solid Phase Extraction (SPE) ... but is not clear whether this is a general treatment or a treatment before compositional analysis, as HPLC or even MS/MS (not shown in the paper but announced...). Anyway, what is the detailed treatment of the sample before enzymatic analysis?
-The acronyms for the different treatments, DW, PMH, AMH and FMH should be defined from the very beginning, but now reader has to wait until lines 425-426 to get them. Besides, references regarding the properties of the enzymes in such treatments and also on related ones should be provided. Probably information about is in the listed references, but they should be linked.
-All figures have the name of the first and last author, and the year, in the beginning, and this should be removed. Besides, the lettering of Figures 2-7 is too small and requires mending.
-Line 118, explain or put reference link for the BrdDU-assay. Line 119, what is "Maca"? Lines 172-174, Muscle Hypertrophy or Hypotrophy?
-Section 2.5, line 187. The kind of HPLC column here used in the analysis should be indicated. According to section 5.9, line 476, it seems that a C18 column has been used for such purpose.
-Lines 194-195 and others (like at Discussion), why authors make emphasis on the proper content on "essential amino acids" in the AMH extracts and hydrolysates? Do they expect that the expected beneficial effects of such extracts/hydrolysates originates from the amino acid composition or better (and specifically) from the generated peptides, from the peptide level? This is an essential reasoning and authors should treat such issue in the work, even if they do not characterise in detail the peptide complement, a very desirable task to perform.
-Section 5.3, Preparation of Hydrolysates. There it is said that the powder homogenate is dissolved in distilled water, adjusted to pH 8.00, enzyme added, and hydrolysis allowed to proceed along 24h, at 55ºC. Are the authors aware that, without the addition of a buffer, most probably the pH of the solution would become acidified along the hydrolysis and, also probably, the added enzyme inactive after a few hours (because of autolysis)? Therefore this is an uncontrolled experiment.
-Line 415, indicated the reference for the MTT assay.
-In the References section is quite usual that references are displayed only with the first author, as it happens in refs. 2,3 and 5. According to the MDPI/Marine Drugs rules, only when the number of authors are higher than 10 such abbreviation can me made, nevertheless keeping the name of the first 10 authors there. This, therefore, should be mended for many of the present references.
Reviewer 3 Report
Comments and Suggestions for Authors
Manuscript ID MARINEDRUGS-2613638, entitled "Alcalase-Assisted Mytilus edulis Hydrolysate Prevents Dexamethasone-Induced Muscle Atrophy In Vitro," is, in my opinion, a well-prepared and presented scientific work. It presents the potential of AMH in the recovery from muscle atrophy. I believe that the content of the manuscript corresponds to the profile of the journal MARINEDRUGS. The manuscript is written in the correct language, properly organized, and divided into chapters. The background was properly defined by the authors. The purpose and scope of the research are clearly stated. The authors used appropriate statistical tools to show the significance of differences between the analyzed variables. In my opinion, the manuscript requires a corrections and additions. My critical remarks below:
1. Introduction needs to be separated into four distinct paragraphs, 1st paragraph: problem statement, 2nd: current solution and limitation, 3rd: proposed solution and 4th: aims and objectives of this research paper.
2. If authors can make more comparisons with other works, it will help readers have a comprehensive understanding of the work. It would be better if authors summarized recently related works and made comparisons. For example, authors can compare the muscle atrophy in vitro with different hydrolysates in a table.
3. The authors should briefly comment on the disadvantages of the studied systems. One whole paragraph should be added to let readers get a better insight into the subject.
4. The entire manuscript needs to be formatted according to the journal guidelines. There are several spacing and other errors, which need to be corrected.